# Red-shifting mutation of light-driven sodium-pump rhodopsin

Keiichi Inoue [1,2,3,4], María del Carmen Marín [5,6], Sahoko Tomida[1], Ryoko Nakamura[1], Yuta Nakajima[1], Massimo Olivucci [5,6,7,8] & Hideki Kandori [1,2]

Microbial rhodopsins are photoreceptive membrane proteins that transport various ions using light energy. While they are widely used in optogenetics to optically control neuronal activity, rhodopsins that function with longer-wavelength light are highly demanded because of their low phototoxicity and high tissue penetration. Here, we achieve a 40-nm red-shift in the absorption wavelength of a sodium-pump rhodopsin (KR2) by altering dipole moment of residues around the retinal chromophore (KR2 P219T/S254A) without impairing its ion-transport activity. Structural differences in the chromophore of the red-shifted protein from that of the wildtype are observed by Fourier transform infrared spectroscopy. QM/MM models generated with an automated protocol show that the changes in the electrostatic interaction between protein and chromophore induced by the amino-acid replacements, lowered the energy gap between the ground and the first electronically excited state. Based on these insights, a natural sodium pump with red-shifted absorption is identified from *Jannaschia seosinensis*.

[1] Department of Life Science and Applied Chemistry, Nagoya Institute of Technology, Showa-ku, Nagoya 466-8555, Japan. [2] OptoBioTechnology Research Center, Nagoya Institute of Technology, Showa-ku, Nagoya 466-8555, Japan. [3] The Institute for Solid State Physics, The University of Tokyo, 5-1-5 Kashiwanoha, Kashiwa, Chiba 277-8581, Japan. [4] PRESTO, Japan Science and Technology Agency, 4-1-8 Honcho, Kawaguchi, Saitama 332-0012, Japan. [5] Dipartimento di Biotecnologie, Chimica e Farmacia, Università di Siena, Via A. Moro 2, I-53100 Siena, Italy. [6] Department of Chemistry, Bowling Green State University, Bowling Green, Ohio 43403, United States. [7] Institut de Physique et Chimie des Matériaux de Strasbourg, UMR 7504 Université de Strasbourg-CNRS, F-67034 Strasbourg, France. [8] USIAS Institut d'Études Avancées, Université de Strasbourg, 5 allée du Général Rouvillois, F-67083 Strasbourg, France. Correspondence and requests for materials should be addressed to H.K. (email: kandori@nitech.ac.jp)

Microbial rhodopsins are photoreceptive membrane proteins widely distributed among diverse eubacteria, archaea, eukaryotic algae, fungi, and giant viruses[1,2]. The first microbial rhodopsin, bacteriorhodopsin (BR), which was identified in *Halobacterium salinarum* in 1971, is a light-driven outward proton (H$^+$) pump[3,4]. Subsequently, other types of microbial rhodopsins with various biological functions were reported, such as the inward chloride (Cl$^-$) pump[5–8], outward sodium (Na$^+$) pump[9,10], inward H$^+$ pump[11], light-gated cation and anion channels[12–14], and enzymatic rhodopsins[15–17]. Ion transporting rhodopsins (pumps and channels) are also most frequently used in optogenetics to optically regulate neuronal activity[18,19].

The optical control using ion-transporting rhodopsins with longer-wavelength light is one of the most important elemental technology for optogenetic application because of the higher penetration depth and lower phototoxicity. To achieve a more ideal optogenetic tool, new types of rhodopsin proteins absorbing longer-wavelength light are being investigated by searching natural rhodopsins[19,20], molecular screening of amino acid mutants and/or chimeric constructs[21,22], and reconstitution with retinal analogues[23–25]. Here we propose a different approach to construct mutants based on structural insights reported for various rhodopsins. By taking account of such insights, it is expected that one is able to adopt a more rational design strategy to reduce the experimental cost of random screening of large number of constructs; additionally, the problem of binding competition between authentic retinal and the artificial analogues in vivo can be avoided.

The absorption maximum wavelength ($\lambda_{max}$) of rhodopsins is determined by the energy gap between the ground (S$_0$) and first electronically excited (S$_1$) state ($\Delta E_{S1-S0}$) of its retinal chromophore (the protonated Schiff-base of retinal). The $\Delta E_{S1-S0}$ value depends on the steric and electrostatic interaction between the chromophore and protein environment. The former regulates the planarity and distortion of the chromophore; e.g. the mutation twisting the retinylidene backbone shortens the π-conjugation system of the chromophore, resulting in the shortening of the $\lambda_{max}$[26,27]. However, because the chromophore of most microbial rhodopsins has higher planarity, it is difficult to achieve further red-shift by increasing it.

In contrast, the regulation of $\Delta E_{S1-S0}$ by electrostatic interaction is related to the electronic structure of the retinal chromophore. While this has a localized positive charge on the Schiff-base linkage in the S$_0$ state, it delocalizes in the S$_1$ state toward the β-ionone ring[28–30]. This difference in the localization of positive charge enables the independent tuning of the energy levels of S$_0$ and S$_1$. Specifically, a negative charge near the Schiff-base region selectively stabilizes the S$_0$ state, whereas the energy level of the S$_1$ state is stabilized by a negative charge located near the β-ionone ring[31]. However, since the Schiff-base region forms part of the ion-conducting pathway of rhodopsins, drastic mutations in that region inhibit the ion-transport function. Hence, here we focused on introducing a negative charge near the β-ionone ring to stabilize the S$_1$ level as below.

For animal rhodopsins, which contain 11-*cis* retinal as the chromophore in the dark state, several colour-regulating residues around the retinal have been identified in previous studies. One of the most well-studied examples is the red-shifting O-H bearing residues around the β-ionone ring in the vertebrate middle and long wavelength sensitive (M/LWS) pigments[32]. Three residues at the positions of 180, 277, and 285 in the human red and green pigments are known to play a decisive role between red (~560 nm) and green (~530 nm) absorbing pigments in this group. Specifically, most of the red-absorbing M/LWS have O-H bearing amino acids at these positions (Ser180/Tyr277/Thr285), whereas

they are replaced by non-OH bearing residues in the green-absorbing M/LWS (Ala180/Phe277/Ala285)[33,34]. The mutational experiment, interconverting these residues between the red- and green-absorbing types, suggested that the suitably oriented dipole of the C-O-H groups have a 20–30 nm red-shifting effect[32,35].

The colour-tuning effect caused by the polarity of amino acid residues around the retinal chromophore has also been well studied in microbial rhodopsins. Sensory rhodopsin I (SRI) is the photoreceptor for the positive photo-taxis toward orange light, and its $\lambda_{max}$ is tuned to the region of 560–590 nm[36]. SRI from *Salinibacter ruber* (*Sr*SRI) has a Cl$^-$ ion bound near the β-ionone ring and it shifts the $\lambda_{max}$ of *Sr*SRI from 542 to 556 nm[37]. The O-H bearing Ser141 and Thr142 neighbouring to the β-ionone ring in BR were also suggested to contribute to the red-shift of the absorption[38–40]. Another example of colour tuning in microbial rhodopsins is reported for channelrhodopsin (ChR). ChR is the most widely used optogenetic tool, and a highly red-shifted ChR is strongly demanded. Naturally occurring red-shifted ChR such as the ChR1 from *Volvox carteri* (VChR1) and the ChR from *Chlamydomonas noctigama* (Chrimson) compared to the ChR2 from *Chlamydomonas reinhardtii* (*Cr*ChR2), the standard optogenetic tool, have been reported as well[19]. Recently, the crystallographic structure of Chrimson was solved, providing molecular insights into its red-shifted absorption[41]. The mutational experiment on the residues around the chromophore revealed that three factors are closely related to the red-shift of Chrimson: characteristic protonation state of the counterion, biased distribution of polar residues around the β-ionone ring, and highly rigid chromophore binding pocket. Among them, here we focus on the polar residue near the retinylidene β-ionone ring and the mutations of Ser223 in Chrimson to Gly (identical to *Cr*ChR2) and Ser220 in a chimeric protein (C1V1, constructed from *C. reinhardtii* ChR1 and VChR1) that resulted in ~10-nm blue-shift[21,41]. This suggests that the red-shifting effect caused by polar amino acid residues around the β-ionone ring is common between the animal and microbial rhodopsins. Homologous red-shift induced by a polar residue is also reported for the BR from *Haloquadratum walsbyi* (*Hw*BR). The mutation of Ser149 in the fifth transmembrane helix (helix-E) to Ala (*Hw*BR S149A) showed a 16-nm blue-shifted absorption compared to the wild-type (WT)[26]. Therefore, the regulation of polarity of the residues around the β-ionone ring is considered to be a highly versatile method to develop new optogenetic tools with red-shifted absorption.

KR2 is an outward sodium-pump rhodopsin (NaR) that actively transports Na$^+$ from the cytoplasm to extracellular milieu, and it was reported to be able to inhibit neuronal spiking as a new type of optogenetic tool without the unnecessary intracellular Cl$^-$ accumulation and pH change, which is sometimes demonstrated to cause unexpected cellular activity[9,10]. However, since it has an absorption in the relatively short wavelength region ($\lambda_{max}$ = 525 nm), colour tuning to a more red-shifted region is required for in vivo applications.

To achieve this, here we attempt to regulate the absorption of KR2 by controlling the polarity of amino acid residues around the chromophore by systematic mutation. As the result, KR2 P219T/S254A in which hydrophilic and hydrophobic residues are introduced near the β-ionone and Schiff-base side of retinal chromophore, respectively, shows 40-nm red-shifted absorption from KR2 WT without impairing Na$^+$-transport efficiency. Fourier transform infrared (FTIR) spectroscopy and quantum mechanics/molecular mechanics (QM/MM) models generated with an Automatic Rhodopsin Modeling (ARM) protocol suggest that structural change of retinal chromophore and the energy gap between the ground and the first electronically excited state is lowered by the change in electrostatic interaction between retinal

chromophore and surrounding amino acid residues. Based on these insights, we identified naturally red-shifted NaR (*Js*NaR) from an α-proteobacterium, *Jannaschia seosinensis*, which shows red-shifted absorption due to the replacement of KR2 Pro219 to Gly.

## Results

**Red-shifted colour variants of KR2.** First, we introduced polar residues in KR2 at homologous positions to those present around the β-ionone ring in Chrimson (Fig. 1 and Supplementary Fig. 1). The visible absorption maxima of the KR2 WT and mutants were determined with bleached spectra upon the hydrolysis reaction of the chromophore of KR2 in the solubilized *Escherichia coli* membrane with hydroxylamine (Fig. 2). The mutants in which the residue is replaced with corresponding amino acid of Chrimson did not show significant absorption shift (KR2 L168Y: 526 nm, KR2 F178Y: 526 nm) or unexpectedly blue-shifted (KR2 G171S: 515 nm). Next, we investigated further nine mutants (KR2 G146S, M149S, I150T, G153S, G156S, I181T, F211Y, W215Y, and

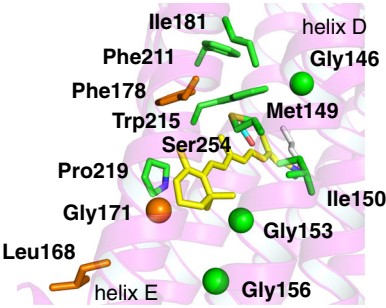

**Fig. 1** Non-polar amino acid residues around the retinylidene β-ionone ring in KR2. The residues mutated to the identical ones as occurring in Chrimson and nine further screened residues are coloured in orange and green, respectively. The Cα atoms are shown as spheres for Gly residues. Ser254 near the retinylidene moiety (in yellow) is coloured in cyan

P219T) in which the hydrophobic residues around the retinylidene β-ionone ring in the crystallographic structure of KR2 were replaced with OH-group bearing one (Fig. 1). While eight of them did not show significant red-shift in the absorption (−16- to +4-nm shift in $\lambda_{max}$), KR2 P219T showed a 17-nm red-shifted absorption ($\lambda_{max}$ = 542 nm) compared to the WT.

For optogenetic applications, a highly efficient transport activity similar to that of WT must be retained even after red-shifting. To confirm this requirement, we assayed the ion-transport activity of KR2 P219T compared to that of WT in *E. coli* cells (Fig. 3). We observed pH increase upon illumination in NaCl solution, representing the secondary $H^+$ uptake compensating the increased membrane potential caused by the outward $Na^+$ transport. While the efficiency of $H^+$ uptake depends on the permeability of the membrane that is modulated by the endogenous membrane proteins in *E. coli*, the addition of protonophore, 10 μM CCCP, makes the membrane highly permeable and the amount of $H^+$ uptake depends only on the change in the chemical potential induced by the $Na^+$ transport. In this case, the extent of $H^+$ uptake more precisely represents the $Na^+$ transport by KR2. Therefore, we estimated $Na^+$ transport activity from the initial slope of pH increase with CCCP (Fig. 3b), and KR2 P219T showed identical $Na^+$ transport activity to that of WT.

These results suggest that KR2 P219T is more ideal as an optogenetic tool compared to KR2 WT (longer-absorption wavelength and identical $Na^+$ transport ability). To achieve further red-shift in KR2, we focused on Ser254, which occurs one residue before the retinylidene-binding Lys255. In several previous studies, the residue in this position has a colour-tuning effect according to the type of the amino acid[26,39,42–44]. Specifically, hydrophobic residues such as Ala have red-shifting effect compared to hydrophilic ones such as Ser and Thr. According to this rule, we constructed the KR2 S254A mutant and it showed a 21-nm red-shifted absorption (Fig. 2) and identical transport activity to that of WT (Fig. 3). This indicates that the S254A mutation enables longer-wavelength absorption without affecting the transport, similar to P219T. Furthermore,

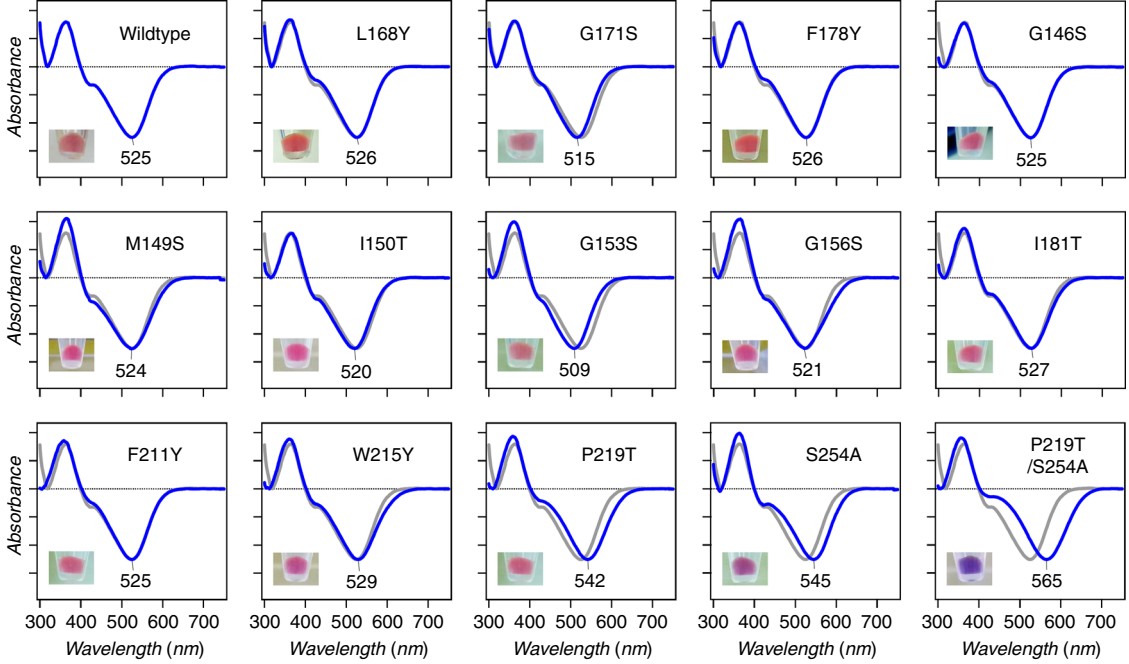

**Fig. 2** $\lambda_{max}$ determination for KR2 mutants. Difference in ultraviolet–visible absorption spectra of the KR2 mutants before and after the bleaching reaction with hydroxylamine and pictures of the pellets of *E. coli* cells expressing the proteins. Source data are provided as a Source Data file

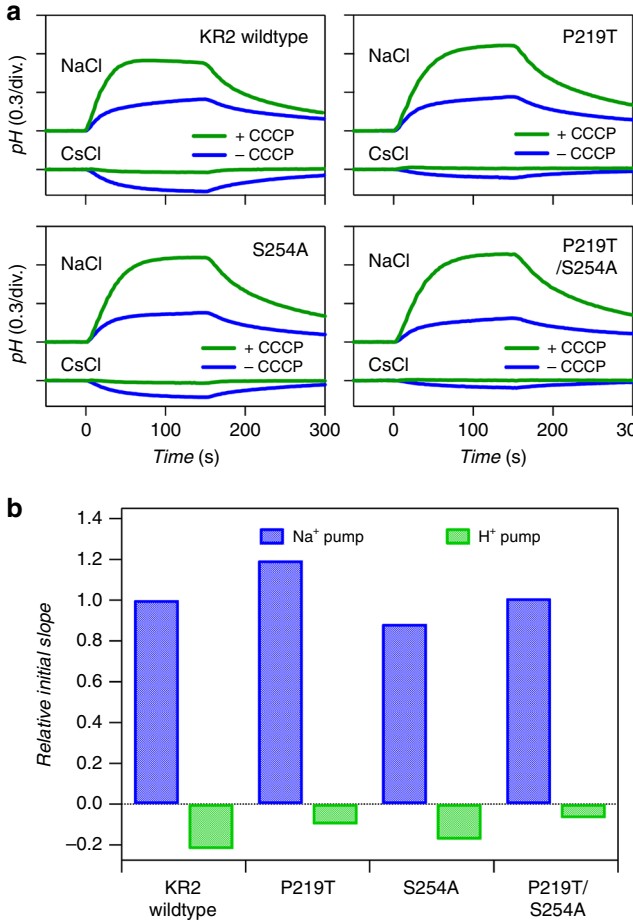

**Fig. 3** Ion pump activity of KR2 mutants. Ion transport activities of the KR2 wildtype (WT) and mutants were assayed by monitoring the pH changes in the external media of the suspension of *E. coli* cells (**a**) and the relative initial slopes of the pH change of KR2 WT, P219T, S254A, and P219T/S254A (**b**). The initial slopes of $Na^+$ and $H^+$ pumps were determined from the results of NaCl + CCCP and CsCl, respectively, shown in **a**. The light was illuminated at $t = 0$-150 s. Source data are provided as a Source Data file

we attempted the combination of them (KR2 P219T/S254A). The double mutant showed >40-nm red-shift from WT with a deep purple colour (Fig. 2) and the transport activity remained unaffected at the same level as the WT (Fig. 3). The colour shift of the P219T/S254A mutation ($\Delta\lambda = +40$ nm equivalent to $-1349$ cm$^{-1}$) is approximately equal to the sum of those of the single mutations (P219T: $-597$ cm$^{-1}$ + S254A: $-699$ cm$^{-1}$ = $-1296$ cm$^{-1}$). This suggests that these two types of mutations on both sides of the retinal chromophore (the β-ionone and Schiff-base side) independently affect its light absorption properties.

**Spectroscopic analysis of KR2 mutants**. The unaffected transport activity of KR2 P219T/S254A suggests that the turnover rate of its photocycle would be similar to that of the WT. To confirm this, we purified the proteins and conducted laser flash photolysis experiment with the sample reconstituted in liposomes of the mixture of 1-palmitoyl-2-oleoyl-phosphatidyl-ethanolamine (POPE) and 1-palmitoyl-2-oleoyl-sn-glycero-3-phosphoglycerol (POPG) (Supplementary Fig. 2). The double mutant showed a photocycle involving the K, K/L, L/M, and O intermediates similar to the WT, but with a turnover rate slower of less than a

factor of 2 (WT: 4.8 ms, P219T/S254A: 8.2 ms) only. The highly similar photo-reaction cycle of the double mutant and WT is consistent with the identical transport activity demonstrated in *E. coli* cells and with the fact that both of the P219T and S254A mutations do not affect the $Na^+$ transport process in the protein.

To gain structural insight into the retinal binding site of the colour-shifted mutants, we perform FTIR spectroscopic measurements and support such an investigation by building and analysing QM/MM models of KR2 WT and its mutants (see section "QM/MM model of KR2 WT and mutants"). The whole spectra in the 1800–800 cm$^{-1}$ range is shown in Supplementary Fig. 3, and the selected regions related to the retinal chromophore are shown in Fig. 4. In the 1225–1155 cm$^{-1}$ region, where the bands for C-C stretching vibrations appear, the 1201($-$), 1194 ($+$), and 1184($-$) cm$^{-1}$ bands were observed in both the WT and the mutants; whereas the band at 1167($-$) cm$^{-1}$ in the WT upshifted to 1174($-$) cm$^{-1}$ in KR2 P219T and P219T/S254A but not in S254A. The bands at 1201($-$), 1194($+$), and 1167($-$) cm$^{-1}$ were previously assigned to the C14-C15 stretch, a mixture of C10–C11 and C14–C15 stretches, and C10-C11 stretch, respectively[45]. The origin of the H/D-sensitive band at 1184($-$) cm$^{-1}$ is not yet known. The specific shift in the band at 1167($-$) cm$^{-1}$ implies that the P219T mutation at the β-ionone side affects the bond order of nearby C10-C11 in the dark state, but not the distant bond (C14-C15). As we will detail in the next section, the higher shift of the 1167($-$) cm$^{-1}$ band in the P219T and P219T/S254A mutants is consistent with the result of QM/MM calculations showing a decrease of the C10-C11 bond distance from 1.46 Å in the WT to 1.45 Å in the mutants (see Fig. 5b and Supplementary Table 3).

In the 1050–880 cm$^{-1}$ region, the hydrogen-out-of-plane (HOOP) modes of the chromophore that are highly informative for the distortion of its backbone were observed (Fig. 4, right). While the 959($+$)-cm$^{-1}$ band did not show a significant shift, the bands at 1010($-$), 984($+$), and 917($-$) cm$^{-1}$ were shifted in the mutants. Because these three vibrations are $D_2O$ sensitive, the shift of their bands represent the changes in the distortion in the Schiff-base region, which were affected not only by the mutation of nearby S254A mutation but also of distant P219T. The 1010 ($-$)-cm$^{-1}$ band shifted to 1005 cm$^{-1}$ for the S254A and P219T/S254A mutants, and not for P219T, suggesting that this band is highly sensitive to the residue at position of 254. The band at 917 ($-$) cm$^{-1}$ is similar to those observed at 911($-$) and 921($-$) cm$^{-1}$ in BR and *Natronomonas pharaonis* phoborhodopsin (also called *pharaonis* sensory rhodopsin II), respectively, which were assigned to the C15-H and N-H coupled HOOP mode[46]. This shift suggests that the structure of the chromophore Schiff-base region is changed in the S254A and P219T mutants. By observing Fig. 5c and Supplementary Table 4, we see that the QM/MM models show an increase in the C11=C12 to C15=N dihedral distortion of both S254A and P219T/S254A mutants relative to the WT form, showing the strongest distortion in the C15=N dihedral angle. A lesser change has instead been in the P219T mutant.

In the 1580–1480 cm$^{-1}$ region, the C=C stretching vibrations of the retinal are observed. The 1535($-$)-cm$^{-1}$ band of KR2 WT shifted to 1533($-$) and 1532($-$) cm$^{-1}$ in the P219T and S254A mutants (Fig. 4, left). The linear correlation between the frequency of C=C str. ($\nu_{C=C}$) and $\lambda_{max}$ is well known, and $\nu_{C=C}$ shows ca. 70-cm$^{-1}$ downshift for the 270-nm red-shifted $\lambda_{max}$[47]. On the basis of this correlation, we expect that the $\nu_{C=C}$ of KR2 P219T (17-nm shifted) and S254A (20-nm shifted) are 4 and 5 cm$^{-1}$ lower than that of the WT form, respectively, and therefore similar to the experimental values (3 and 4 cm$^{-1}$, respectively). The corresponding band of KR2 P219T/S254 showed larger downshift to 1525($-$) cm$^{-1}$. Unexpectedly, the intensity of the C=C str. bands of the KR2 S254A and

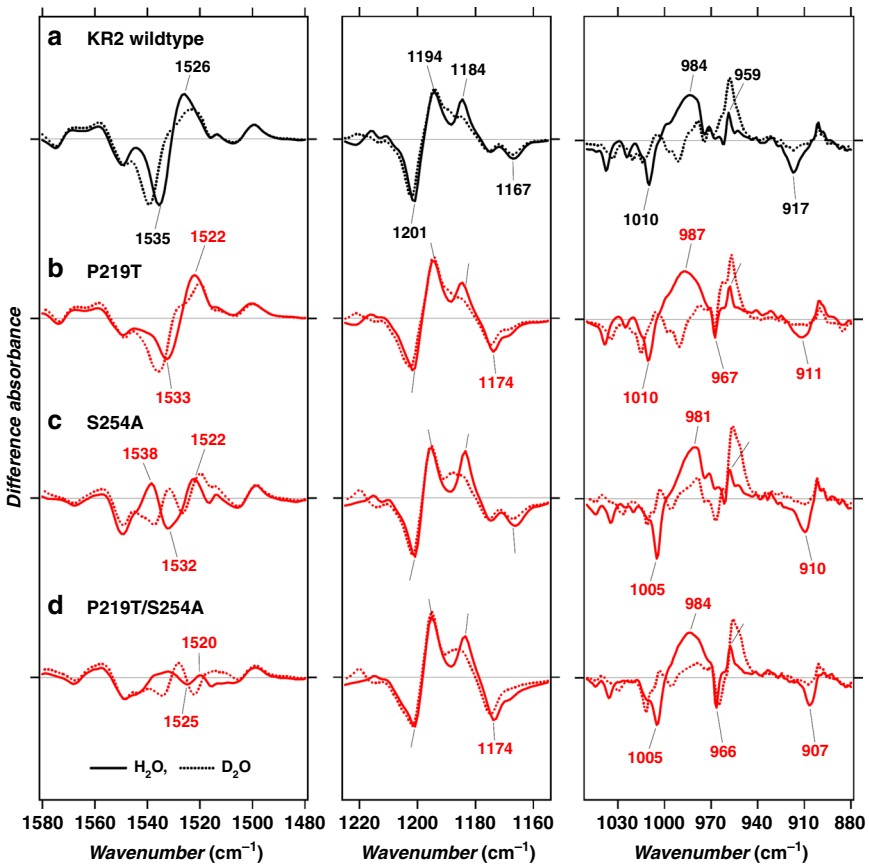

**Fig. 4** Light-induced infrared absorption changes of the KR2 mutants. Light-induced Fourier transform infrared difference spectra of **a** KR2 wildtype (black) and the mutants (red), **b** P219T, **c** S254A and **d** P219T/S254A, in the 1580–1480 (left), 1225–1155 (middle) and 1050–880 cm$^{-1}$ (right) regions at $T = 77$ K and pH 8.0. Solid and dotted lines represent the samples hydrated with $H_2O$ and $D_2O$, respectively. Source data are provided as a Source Data file

P219T/S254A mutants was significantly lower than that of KR2 WT and P219T, suggesting that the mutation S254A affects the extinction coefficient ($\varepsilon$) of the C=C str. band. Hence, as we described above, the shift in $\lambda_{max}$ is related to the delocalization of $\pi$-electron. If the $\pi$-electron is more delocalized, the bonding order of C=C decreases and the C=C stretching bands downshift, and vice versa. In our case, once the double bonds get conjugated, the system is getting longer and hence the wavelength is red-shifted. One effect of the conjugation is that the double bonds get a little bit longer, and the in between single bonds get a little bit shorter. So, more conjugation implies less bond length alternation (BLA) and red-shift. In this way, the shift of the 1535 (−)-cm$^{-1}$ band observed in the different mutants with respect to the WT is due to an increase in the C=C bond distance. Therefore, the mutant with the highest downshift will have the lowest BLA and its double bonds will be longer with respect to the rest of the models. The QM/MM models (Fig. 5b, top) corroborate this hypothesis showing in the double mutant (P219T/S254A) the longest double bonds with respect to the WT and mutants and also the lowest BLA value (WT 0.1160, P219T 0.1123, S254A 0.1120, and P219T/S254A 0.1093). Notice that the increase in conjugation discussed above must be due to an electrostatic effect. In fact, by stabilizing the positive charge in the β-ionone ring region one also stabilizes a resonance formula with inverted single and double bonds along the chromophore backbone (see details in Supplementary Table 5).

**QM/MM models of KR2 WT and its mutants**. As anticipated above, in order to get further insight into the mechanism of the absorption red-shifting, we have built suitable QM/MM models

for KR2 WT and its P219T, P219G, S254A, and P219T/S254A mutants constructed by taking the WT X-ray crystallographic structure (PDB code: 3X3C)[10] as the template. The "closed" conformation (rotamer 2) of residue Asp116 has been selected on both the basis of crystallographic data as well as for consistency with the proposed pumping mechanism[10]. However, an "open" (rotamer 1) conformation is also possible as suggested by the same crystallographic data. Such models were generated using the ARM protocol[48] (see "Methods"), which also compute the vertical excitation energy ($\Delta E_{S1-S0}$) between the ground ($S_0$) and first electronically excited ($S_1$) singlet states. These values can then be compared with the corresponding experimental result derived from the observed $\lambda_{max}$. As previously reported for other microbial rhodopsins[48], we have found that the computed $\Delta E_{S1-S0}$ values feature a few kcal mol$^{-1}$ systematic blue-shifted error. However, most importantly, the trend of the energy shifts of the mutants relative to the WT form were reproduced thus supporting the validity of the models (Table 1 and Fig. 5a).

The constructed QM/MM models allowed to disentangle the electrostatic and steric effects responsible for the observed $\Delta E_{S1-S0}$ red-shifted values. To do so, we computed the $\Delta E_{S1-S0}$ values of the retinal chromophore in isolated condition (i.e. removing the protein part from the model while keeping the chromophore geometry fixed at its equilibrium geometry in the protein environment). The changes in such gas-phase $\Delta E_{S1-S0}$ values relative to their WT reference reflect the changes in the chromophore geometry induced by the mutation and are mainly assigned to steric interactions. The difference between the total and vacuum $\Delta E_{S1-S0}$ values is instead assigned to electrostatic interactions (Table 1). It can be observed that, as previously

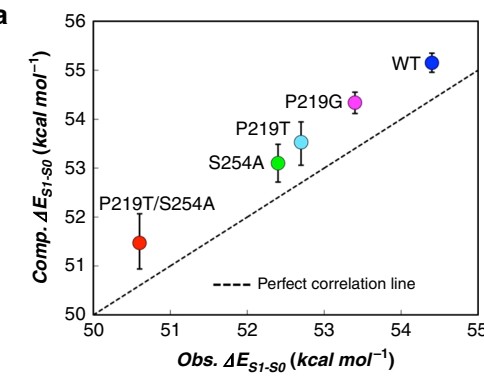

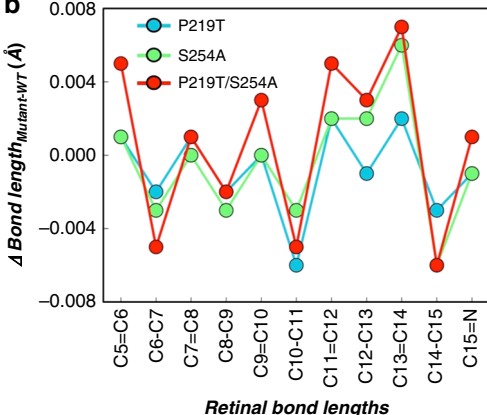

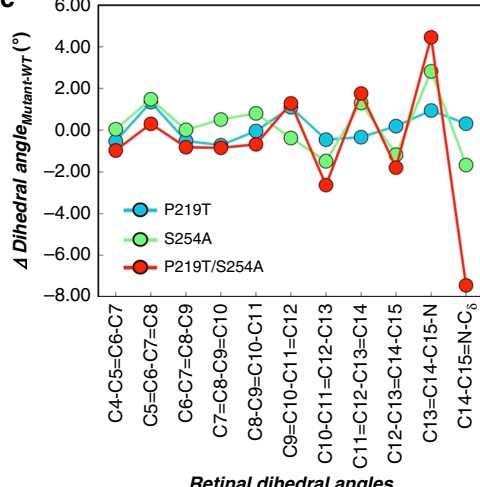

**Table 1 Vertical excitation energies of the retinal chromophore incorporated in the protein and in vacuum**

| Protein | $\Delta E_{S1-S0}$ (Protein) (kcal mol$^{-1}$) | $\Delta E_{S1-S0}$ (Vacuum) (kcal mol$^{-1}$) | $\Delta E_{S1-S0}$ (Protein − Vacuum) (kcal mol$^{-1}$) |
|---|---|---|---|
| KR2 WT | 55.2 | 43.1 | +12.1 |
| P219G | 54.3 (−0.9) | 43.8 (+0.7) | +10.5 (−1.6) |
| P219T | 53.5 (−1.7) | 44.5 (+1.3) | +9.0 (−3.1) |
| S254A | 53.1 (−2.1) | 43.6 (+0.5) | +9.5 (−2.6) |
| P219T/ S254A | 51.5 (−3.7) | 45.9 (+2.7) | +5.6 (−6.5) |

The energy differences ($\Delta E_{S1-S0}$) between the ground ($S_0$) and first electronically excited state ($S_1$) were calculated by the QM/MM models using the ARM protocol. The values for the retinal chromophore in the protein (Protein), isolated in vacuum (Vacuum), and their difference (Protein − Vacuum) are shown. The values in the parenthesis for the mutants show the difference from KR2 WT

the retinal geometry cause a blue-shift with respect to the WT. However, as shown in Table 1 (right column), we also found that a red-shifting electrostatic contribution imposed by the protein environment dominates, thus quenching the chromophore steric effect and resulting in a net red-shifted change, with the strongest effect observed for the P219T/S254A mutant. In conclusion, the QM/MM analysis reveal that the observed increase in $\lambda_{max}$ values along the WT KR2, P219T, P219G, S254A and P219T/S254A series is dominated by the change in the electrostatics of the chromophore-hosting cavity. In Fig. 6, we provide a qualitative explanation of such electrostatic effects, which shows how the P219T mutation creates a new dipole moment having the partially negatively charged oxygen atom of the Thr side chain facing the β-ionone ring, thereby selectively stabilizing the $S_1$ relative to $S_0$. In contrast, S254A mutation removes a dipole moment vector, which stabilizes the $S_0$ relative to $S_1$.

Using the same QM/MM models, it is also possible to investigate the role played by each protein amino acid residues in the described red-shifting relative to the WT form. In fact, one can set the point charges of each specific residue to zero and then re-compute the $\Delta E_{S1-S0}$ value ($\Delta E_{off}$). After examining the changes in $\Delta E_{off}$ values of different residues in the mutant and in the WT, it can be observed that the change in $\Delta E_{off}$ may have two components: (i) the first is a direct component due to a change in number, magnitude, and position of the point charges in the mutated site and (ii) the second component is indirect and originates from the reorganization of the local environment and hydrogen bond network induced by the same mutation. The second effect is due the fact that, in the mutants, conserved residues and water molecules change in position or conformation thus displaying different $\Delta E_{S1-S0} - \Delta E_{off}$ values with respect to the WT. Such effects contribute to the total $\Delta E_{S1-S0}$ changes significantly. For instance, when we compare the single mutants (P219T, P219G and S254A), the double mutant (P219T/S254A), and the WT, the data show that the amino acid substitutions at residues 219 and 254 result in a red-shift in absorption relative to the WT and directly contribute to the strong red-shifting observed in the double mutant (see Supplementary Table 2). However, such changes are accompanied by variations in the effects of the conserved residues. In fact, the effect of the conserved Ser254 residue in P219T and the conserved Pro219 residue in S254A are not the same in the WT form and in the mutants.

**Natural NaR lacking homologous proline to KR2 Pro219**. In this study, we revealed that the mutation at Pro219 in KR2 affects the absorption wavelength of the chromophore. On the basis of this insight, we explored NaRs lacking proline at homologous

**Fig. 5** Excitation energy and conjugated structure of retinal chromophore computed by quantum mechanics/molecular mechanics (QM/MM) models. **a** Comparison between the computed and observed vertical excitation energies, $\Delta E_{S1-S0}$ (kcal mol$^{-1}$) of QM/MM models built with ARM protocol at CASPT2//CASSCF(12,12)/6-31G*/AMBER level of theory for KR2 wildtype (WT) (dark blue), P219G (pink), P219T (clear blue), S254A (green), and P219T/S254A (red) mutants. The error bars of the standard deviation are shown in black (see details in Supplementary Table 1). **b** Bond lengths and **c** chromophore dihedral angle differences of each mutant relative to KR2 WT at CASSCF(12,12)/6-31G*/AMBER level of theory

reported[49], the chromophores extracted from their protein environments feature $\Delta E_{S1-S0}$ values (see $\Delta E_{S1-S0}$ Vacuum column in Table 1) 6–12 kcal mol$^{-1}$ lower than those of the corresponding protein (see $\Delta E_{S1-S0}$ Protein column in Table 1). Interestingly, it was found that the mutation-induced changes in

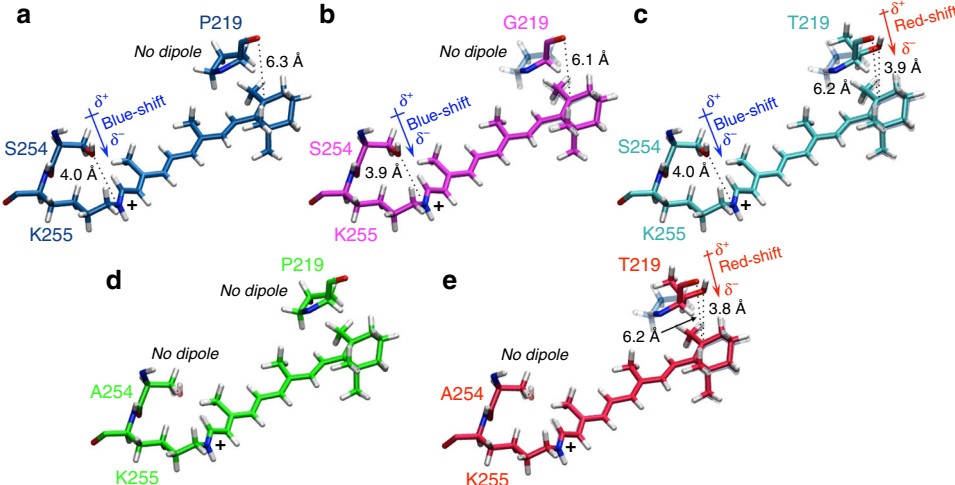

**Fig. 6** Quantum mechanics/molecular mechanics structures around retinal chromophore in KR2 wildtype (WT) and mutants. Comparison between retinal chromophores and mutated residues 219 and 254 in **a** KR2 WT, **b** P219G, **c** P219T, **d** S254A and **e** P219T/S254A mutants. For mutants are also shown, in transparent representation, the retinal chromophore and 219 and 254 residues of KR2 WT

position, which are expected to have red-shifted absorption in nature. This proline is highly conserved in most of the microbial rhodopsins[1]. However, we identified two NaRs that do not have this proline: NaRs from *Parvularcula oceani*[50] (*Po*NaR) and *J. seosinensis*[51] (*Js*NaR) that have Thr and Gly at the position of KR2 Pro219, respectively. Although the former was reported in our previous study, it was not expressed in *E. coli* cells[11]. In contrast, we were able to express *Js*NaR in *E. coli* cells, which had purple colour similar to KR2 P219T, suggesting that it has red-shifted spectrum compared to the KR2 WT (Fig. 7a, left). We performed the hydroxylamine bleaching experiment for *Js*NaR, and it showed 25-nm longer $\lambda_{max}$ (550 nm) compared to the KR2 WT (Fig. 7a, right). *Js*NaR works as a Na$^+$ and H$^+$ pump in NaCl and CsCl solutions (Fig. 7b), respectively, similar to KR2[11]. Transient absorption measurement also showed a photocycle similar to that of KR2[11] (Fig. 7c). Hence, the difference between *Js*NaR and KR2 is only in their absorption wavelength.

In order to confirm that the difference in amino acid at the position of KR2 Pro219 was solely responsible for the red-shifted $\lambda_{max}$ of *Js*NaR, we mutated the corresponding residue of *Js*NaR, Gly216, to Pro, and the mutant showed identical $\lambda_{max}$ (523 nm) to that of KR2 WT (Fig. 7a). However, when we introduced the S247A mutation (homologous to KR2 S254A), the $\lambda_{max}$ was further red-shifted to 569 nm. While the dipole moment, induced by the O atom in the side chain of Thr219, largely contributed to the selective stabilization of the S$_1$ state in KR2 P219T, Gly216 of *Js*NaR is a non-polar residue without the O-H group. Hence, another mechanism is considered to affect the red-shift. For neutral amino acids such as glycine, there is a dipole that is originated from the C=O and N-H groups of the backbone and it might have a red-shifting effect if a change in this dipole occurs after the mutation. Identical KR2 mutant (KR2 P219G) also showed a 10-nm red-shift ($\lambda_{max}$ = 535 nm, Fig. 7a, right), and the shift was reproduced by the QM/MM model using ARM (Fig. 5a). The modelled structure shown in Fig. 6, in which the negative charge of the dipole moment formed by the C=O and N-H groups of the backbone is oriented towards the β-ionone ring, together with the point charge analysis described above provide an explanation of the observed red-shift.

### Discussion
To achieve an ideal optogenetic tool for in vivo studies, red-shifting colour tuning of rhodopsin is highly demanded. In this

study, we have initially attempted to shift the colour shift of KR2 by introducing OH-group bearing amino acid residues in the β-ionone ring region according to their position in Chrimson (L169Y, F178Y, and G171S). However, such attempt did not yield the desired result as the KR2 mutation showed no significant absorption shift or they rather caused an unexpected blue-shift (Fig. 2). This suggests that, unlike in Chrimson, these residues are not involved in the control of the spectrum of KR2. This might be due to the difference in the interaction between the introduced and the surrounding residues. When we compare the crystal structure of KR2 (PDB code: 3X3C)[10] and Chrimson (5ZIH)[41], half of the residues (5 out of 11) within 5 Å from the β-ionone ring are different between the two proteins. The residues in Chrimson are reported to determine a tightly packed cavity that increases the protein rigidity around the retinal chromophore also leading to the increase in the planarity of the chromophore with a consequent spectral red-shift[41]. Thus the differences in the rigidity of the protein around the β-ionone ring may be responsible for inducing the different colour-tuning effects observed between KR2 and Chrimson. The role of protein rigidity in colour tuning is not well understood compared to the role of amino acid polarity, and it should be more comprehensively studied in future work.

In order to red-shift the KR2 absorption, we then screened nine mutations of hydrophobic residues around the retinal β-ionone ring. As a result, the mutation of KR2 Pro219 to Thr yielded a significantly 17 nm absorption red-shift. Furthermore, the double mutant, KR2 P219T/S254A, showed further red-shifted $\lambda_{max}$ (+40 nm) with identical Na$^+$ transport activity to that of the WT.

The identical transport activity and photocycle of the mutants involving P219T and S254A suggest that they are not directly related to the ion-transport function of KR2. In contrast, FTIR spectroscopy and QM/MM modelling indicated that the mutations affect the distortion of the retinal but in a direction yielding a blue-shift due to an increase of $\Delta E_{S1-S0}$. However, our QM/MM analysis shows that the change in the protein environment, especially the changes in the dipoles at 219 and 254 positions make the gap much smaller, reproducing the experimental $\lambda_{max}$ shift. This implies that the way to rationally design further red-shifted proteins is by modifying the dipole moments of other residues and/or introducing other types of mutations to recover the chromophore planarity lost in KR2 P219T/S254A.

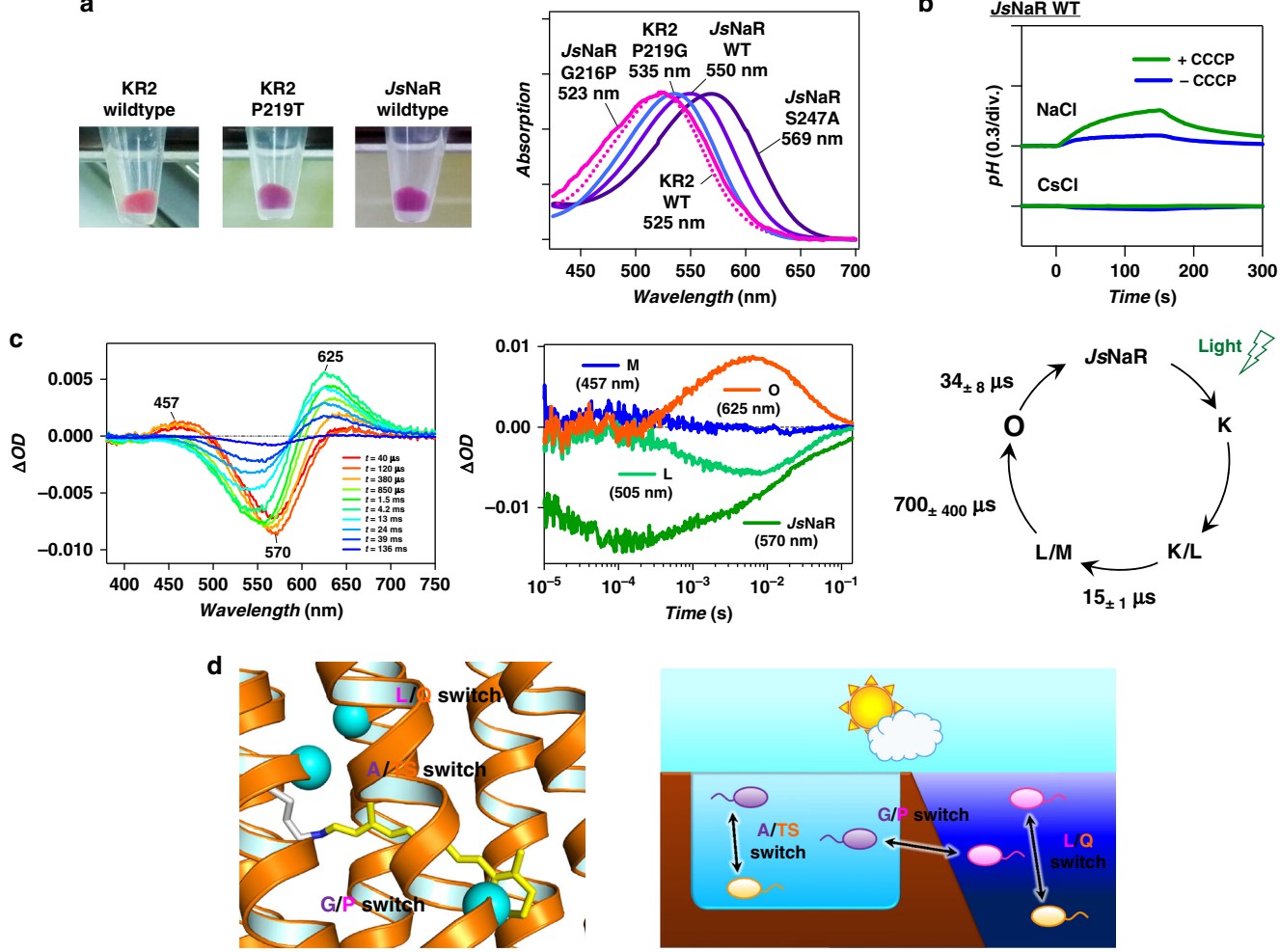

**Fig. 7** The natural red-shifted NaR from *J. seosinensis* (*Js*NaR) without proline residue. **a** Pictures of the pellets of the *E. coli* cells expressing the KR2 wildtype, P219T, and *Js*NaR wildtype (left) and the absorption spectra of *Js*NaR wildtype (purple solid line), G216P (magenta solid line), and S247A mutants (indigo solid line). The absorption spectrum of KR2 wildtype and P219G mutant are shown by the magenta dotted line and cyan solid line, respectively. **b** Ion pump activity assays of the *Js*NaR wildtype. **c** The transient absorption spectra (left), time evolutions of the transient absorption change at specific wavelengths (middle) and the photocycle (right) of the *Js*NaR wildtype. The lifetimes and their standard deviations of the photo-intermediates in the photocycle of *Js*NaR are indicated. Source data are provided as a Source Data file. **d** The residues for colour regulating switches in microbial rhodopsins indicated by cyan spheres (left) and three types of switch working in various bacteria and archaea living in various environments in nature

Since the proline residue homologous to KR2 Pro219 is conserved among most of the microbial rhodopsins, identical Pro-to-Thr mutation could be used to systematically shift the absorption of various types of rhodopsins to longer wavelength. In order to confirm this, we applied similar mutation to the light-driven chloride pump, *N. pharaonis* halorhodopsin (*Np*HR), and the inward H$^+$ pump, *P. oceani* xenorhodopsin (*Po*XeR). Despite both of them having longer $\lambda_{max}$ even in their WT (*Np*HR: 576 nm, *Po*XeR: 569 nm) variant than that of KR2, we observed a 7- and 6-nm further red-shift (*Np*HR P226T: 583 nm, *Po*XeR P179T: 575 nm, Supplementary Fig. 4a), suggesting wide applicability of the mutation of the homologous proline for colour tuning. However, while the *Np*HR mutant showed identical transport activity, the activity of *Po*XeR P179T was about one-fourth of that of the WT (Supplementary Fig. 4b). This result suggests that Pro179 is involved in the inward H$^+$ transport of *Po*XeR but not in KR2 and *Np*HR. Thus the effect of the mutation of the proline on the function needs to be carefully investigated for each rhodopsin type.

As a NaR lacking Pro at the position of KR2 Pro219 in nature, we identified *Js*NaR with a red-shifted spectrum. The phylogenic

position of *Js*NaR in the NaR sub-family is shown in Supplementary Fig. 5, and it is surrounded by proline-bearing NaRs. The result of *Js*NaR and its G216P mutant suggested that the Pro to Gly mutation is related to a natural evolutionary process leading to the red-shifted absorption. To understand the occurrence of red-shifted absorption wavelength in *Js*NaR, we considered the difference in the environment inhabited by *Js*NaR and KR2. *J. seosinensis* was isolated from the hypersaline water of a solar saltern where the water is expected to be more turbid than that in the ocean. Since turbid water scatters shorter wavelength light and the organism living there can use only relatively longer wavelengths, the visual system of animals is optimized to function in the red region[52]. Hence, we currently hypothesize that in *Js*NaR the Pro is replaced to Gly to utilize longer-wavelength light in hypersaline water.

For the proteorhodopsin group, the difference between Leu and Gln at position 105, referred to as the L/Q switch, is well known to regulate the spectra between the green- and blue-absorbing types, respectively[53–55]. This is considered to be related to the optimization of the absorption wavelength to the light region that can be utilized by bacteria living in each depth: specifically, the

colour regulation enables bacteria living in the shallow and deep oceans to use the green and blue light, respectively, which is abundant in their respective habitats (Fig. 7d). Alternatively, a blue-shifting has been associated with the decrease in thermal noise of rhodopsins in general[56] and therefore to the increase in their light sensitivity. This suggests that proteorhodopsins thriving in dim-light environments such as in the deep ocean habitat would tune their absorption maximum to a blue colour.

The residue at the position of BR Ala215 is known to partly contribute to the spectral difference between the BR and *N. pharaonis* sensory rhodopsin II (*Np*SRII, also *pharaonis* phoborhodopsin), it is replaced with Thr in *Np*SRII[39]. This is considered to be related to the evolution from BR to *Np*SRII[57], and the current study indicates that the Ser254 of KR2 also has a similar role. Hence, here we refer to it as the A/TS switch (Fig. 7d), as another example of naturally occurring colour-determining residue. The results for *Js*NaR implies that the red-shifting effect of G216 is related to the colour tuning of the NaR in clear ocean and turbid solar saltern environments (G/P switch in Fig. 7d).

In this study, we obtained the insights that the absorption of the retinal in protein can be red-shifted by properly placing and orienting the dipoles of surrounding residues without affecting the ion-transport activity. The molecular-level design of rhodopsins regulating the absorption wavelength of the retinal chromophore without impairing the transport activity is still difficult compared to the prediction of the only absorption wavelength. Thus our findings expand the experimental basis useful to establish the artificial design of functional molecules useful for optogenetics application. We expect this approach would provide the basis for the development of red-shifted rhodopsins and more ideal optogenetic tools for in vivo applications.

## Methods

**Mutagenesis and protein expression**. The synthesized genes of KR2 (Protein Accession number: BAN14808.1) and *Js*NaR (Protein Accession number: WP_055664831) codon-optimized for *E. coli* were incorporated into the pET21a(+) vector (Novagen, Merck KGaA, Germany). The site-directed mutation was conducted using a QuikChange Site-directed Mutagenesis Kit (Agilent, CA), and the sequences of primers used in mutagenesis are listed in Supplementary Table 6. The plasmids carrying the genes of the KR2 WT, *Js*NaR WT, and mutants were transformed into the *E. coli* C43(DE3) strain (Lucigen, WI). The protein expression was induced with 1 mM isopropyl β-D-1-thiogalactopyranoside in the presence of 10 μM all-*trans* retinal for 4 h.

**Measurement of $\lambda_{max}$ by hydroxylamine bleach**. The $\lambda_{max}$ of KR2 WT, *Js*NaR, and their mutants was determined by bleaching the protein with hydroxylamine according to the previously reported method. The *E. coli* cells expressing the rhodopsins were washed with a solution containing 100 mM NaCl, 50 mM Na$_2$HPO$_4$ (pH 7) for three times. The washed cells were treated with 1 mM lysozyme for 1 h and then disrupted by sonication. To solubilize the rhodopsins, 3% DDM was added and the samples were stirred for overnight at 4 °C. The rhodopsins were bleached with 500 mM hydroxylamine and subjected to illumination of yellow light ($\lambda > 500$ nm) from the output of 1 kW tungsten–halogen projector lamp (Master HILUX-HR, Rikagaku) through a glass filter (Y-52, AGC Techno Glass). The absorption change upon the bleaching was measured by ultraviolet–visible spectrometer (V-730, JASCO, Japan).

**Ion-transport assay**. To assay the ion-transport activity assay in *E. coli* cells, the cells carrying the expressed rhodopsin were washed for three times and resuspended in unbuffered 100 mM NaCl or CsCl to assay the Na$^+$ or H$^+$ pump activity, respectively. The light was illuminated after adjusting the pH to ~7 by the addition of a small amount of HCl or NaOH. The pH change upon light illumination was monitored with a pH electrode (9618S-10D, HORIBA, Japan). The wavelength of the illuminating light was changed by placing different colour filters (Y-52, Y-54, O-55, and O-56, AGC Techno Glass) with a heat-absorbing filter (HAF-50S-50H, SIGMAKOKI, Japan) in front of the light source (1 kW tungsten–halogen projector lamp, Master HILUX-HR, Rikagaku) for KR2 WT ($\lambda > 500$ nm), P219T ($\lambda > 520$ nm), S254A ($\lambda > 520$ nm), P219T/S254A ($\lambda > 540$ nm), and *Js*NaR WT ($\lambda > 540$ nm) to correct for the change in the degrees of light absorption by their spectral shift.

**QM/MM models**. As anticipated above, the QM/MM models of the KR2 WT and its mutants P219T, P219G, S254A, and P219T/S254A have been constructed employing the ARM protocol, which has been benchmarked with respect to a set of animal and microbial rhodopsins from different organisms and a set of mutants from bovine rhodopsin[48]. While ARM models are basic gas phase and globally uncharged monomer models constructed starting from an X-ray crystallographic structure or comparative model, the benchmark shows that ARM models reproduce the experimental $\lambda_{max}$ values with a few kcal mol$^{-1}$ discrepancy and that can reproduce trends in $\lambda_{max}$. Accordingly, in the present contribution, the rhodopsin models are used to study the molecular mechanics determining the $\lambda_{max}$ changes after having been validated by reproducing the observed $\lambda_{max}$. Thus ARM does not reproduce the most accurate QM/MM models possible but computationally fast models for the rationalization of trends between sequence variability and function. We notice that a more realistic but computationally demanding QM/MM model of KR2 has been recently reported[58].

The construction of ARM models is based on the S$_0$ geometry optimization of the model guess structure where the retinal chromophore is treated at the QM level of theory and the corresponding protein cavity residues are treated at the MM level (the protein backbone and residues not belonging to the chromophore cavity are not relaxed). To relax the model geometry, ARM employs the multi-configurational complete active space self-consistent field (CASSCF)[59,60] theory to treat the QM subsystem (single-root CASSCF(12,12)/6-31G* level) while the MM subsystem is treated using the AMBER force field[61]. The $\Delta E_{S1-S0}$ energy is then computed using a suitable multi-configurational second-order perturbation theory (CASPT2)[29,62–64] level to recover the missing dynamical electron correlation associated with the CASSCF description (three-root stage average single point CASPT2/6-31G*/AMBER calculation, with the three-root stage-average CASSCF (12,12)/6-31G* wavefunction as the zero-order wavefunction a reference).

The workflow of ARM starts by reading the PDB file containing an initial (guess) rhodopsin structure. First, using the DOWSER program, the hydrogen atoms are added and their positions are optimized at the MM level (the MM force field Amber94 is used in all calculations). Then a set of $N$ molecular dynamic (MD) room-temperature relaxations (1 ns) is performed at the MM level using Gromacs 4.5.4 (i.e., starting with $N$ different seeds that provide $N$ independent sets of initial velocities). The $N$ output structures constitute the initial guess for $N$ corresponding single point QM/MM calculations. More specifically, $N$ QM/MM geometry optimizations are carried out at the HF/3-21G/MM level leading to the conclusion of the preparatory phase. The $N$ independent QM/MM structures obtained from the MD are relaxed via two sequential QM/MM optimizations. The first is performed at the single-root CASSCF(12,12)/3-21G/MM and the second using a CASSCF(12,12)/6-31G*/MM levels ultimately getting $N$ final structures. Subsequently, a CASPT2(12,12)/6-31G*/MM three-root state-average single point calculation is performed for each structure, taking a three-root CASSCF(12,12)/6-31*/MM wavefunction as a reference. Finally, $N$ vertical excitation energy values are computed as the difference between the energies of the first two roots. The average value is then compared with the corresponding experimentally observed maximum absorption wavelength. All QM/MM calculations are performed with the distributed Molcas 8.1/Tinker 6.3 interface. ARM can optionally perform amino acid substitutions on the starting PDB structure to generate ARM models of rhodopsin mutants. Mutations are carried out by SCWRL4 program, which predict side-chain conformations from a given protein backbone, using a backbone-dependent rotamer library. Once the mutations have been made, ARM workflow proceeds exactly as described above.

**Light-induced low-temperature FTIR spectroscopy**. Low-temperature light-induced difference FTIR spectroscopy was performed as described previously[65,66]. The purified proteins of KR2 P219T, S254A, and P219T/S254A were reconstituted into a mixture of POPE and POPG membranes (molar ratio = 3:1) with a protein-to-lipid molar ratio of 1:50 by removing DDM with Bio-Beads (SM-2, Bio-Rad). The reconstituted samples were washed three times with 1 mM NaCl and 2 mM Tris-HCl (pH 8.5). The pellet was resuspended in the same buffer, where the concentration was adjusted to make the intensity of amide I ~ 0.7. A 60-μL aliquot was placed onto a BaF$_2$ window and dried gently. The films were then rehydrated with 1 μL H$_2$O or D$_2$O, and the sample was placed in an Oxford DN-1704 cryostat mounted in the Bio-Rad FTS-40 spectrometer. For the formation of the K intermediate, samples of P219T and P219T/S254A were illuminated with the 540-nm light (interference filter) from a 1-kW halogen-tungsten lamp for 2 min at 77 K. The K intermediate was photo-reversed with $\lambda > 610$ nm light (R-63 cut-off filter, Toshiba) for 1 min, followed by illumination with the 540-nm light. For the formation of the K intermediate, the sample of S254A was illuminated with the 500-nm light (interference filter) from a 1-kW halogen–tungsten lamp for 2 min at 77 K. The K intermediate was photo-reversed with $\lambda > 610$ nm light (R-63 cut-off filter, Toshiba) for 1 min, followed by illumination with 500-nm light. For each measurement, 128 interferograms were accumulated with 2 cm$^{-1}$ spectral resolution, and 21–44 identical recordings were averaged for each measurement.

**Laser flash photolysis**. For the laser flash photolysis measurement, rhodopsin was purified with a Co$^{2+}$-NTA column (TALON, Qiagen) and reconstituted into a mixture of POPE (Avanti Polar Lipids, AL) and POPG (sodium salt, Avanti Polar Lipids, AL) (molar ratio = 3:1) with a protein-to-lipid molar ratio of 1:50. The

absorption of the protein solution was adjusted to 0.8–0.9 (total protein concentration ~0.15 mg mL$^{-1}$) at the excitation wavelength (532 nm). The sample was illuminated with a beam of second harmonics of a nanosecond-pulsed Nd$^{3+}$-YAG laser ($\lambda = 532$ nm, 3 mJ pulse$^{-1}$, 2 Hz, INDI40, Spectra-Physics, CA). The transient absorption spectra were obtained by monitoring the intensity change of white light from a Xe-arc lamp (L9289-01, Hamamatsu Photonics, Japan) passed through the sample with an ICCD linear array detector (C8808-01, Hamamatsu, Japan). To increase the signal-to-noise (S/N) ratio, 90 spectra were averaged and the singular value decomposition analysis was applied. To measure the time evolution of transient absorption change at specific wavelengths, the light of Xe-arc lamp (L9289-01, Hamamatsu Photonics, Japan) was monochromated by monochromators (S-10, SOMA OPTICS, Japan) and the change in the intensity after the photo-excitation was monitored with a photomultiplier tube (R10699, Hamamatsu Photonics, Japan) equipped with a notch filter (532 nm, bandwidth = 17 nm, Semrock, NY) to remove the scattered pump pulse. To increase S/N ratio, 50–100 signals were averaged.

**Reporting summary**. Further information on research design is available in the Nature Research Reporting Summary linked to this article.

## Data availability

Data supporting the findings of this manuscript are available from the corresponding author upon reasonable request. A reporting summary for this article is available as a Supplementary Information file. The source data underlying Figs. 2, 3a, b, 4a–d, and 7a–c and Supplementary Figs. 2a, b, 3a–d, and 4a, b are provided as a Source Data file.

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

## Acknowledgements

We appreciate Professor Osamu Nureki, Professor Ryuichiro Ishitani, Professor Tomohiro Nishizwa, Mr. Kazumasa Oda, and Mr. Tatsuya Ikuta for the useful discussion on the design of mutational colour tuning. We thank Mr. Shinya Sugita for the purification of proteins. This work was supported by JSPS KAKENHI, Japan, Grant Numbers 26708001, 26620005, and 17H03007 to K.I. and 25104009 and 15H02391 to H.K. and by JST, PRESTO, Japan, Grant Number JPMJPR15P2 to K.I. M.O. is partially funded by the grants NSF CHE-CLP-1710191 and NIH GM126627 01 and a USIAS 2015 grant. M.d.C. M. and M.O. are grateful to MIUR for a Department of Excellence Grant.

## Author contributions

K.I., M.d.C.M, M.O. and H.K. contributed to the study design. K.I., R.N. and Y.N. contributed to the construction of mutant proteins and their introduction into *E. coli*. R.N. conducted the pump activity measurement and determination of $\lambda_{max}$. S.T. measured the FTIR spectra. K.I. conducted the transient absorption measurement, analysis of the kinetic data and phylogenetic analysis. M.d.C.M. constructed the QM/MM models generated with ARM protocol and calculated $\Delta E_{S1-S0}$. K.I., M.d.C.M., M.O. and H.K. wrote the paper. All authors discussed and commented on the manuscript.

## Additional information

**Competing interests:** The authors declare no competing interests.

