## [Peer Review File · Nature Communications]

Reviewers' comments:

Reviewer #1 (Remarks to the Author):

The authors report on the engineering of a red-shifted Na rhodopsin. Super important as a tool for optogenetics.

Remarks:

The first sentence of the introduction should be in plural. RhodopsinS.

It would be extremely important to mark the P219 S254 positions in figure S1. And to also include it that figure the sequence of JsNaR.

Oded Beja

Reviewer #2 (Remarks to the Author):

The manuscript "Red-shifting Mutation of Light-driven Sodium Pump Rhodopsin" by Inoue et al. is describing a 40-nm red-shift of the wavelength of the maximum of absorption (λ_{max}) of a sodium pump rhodopsin (KR2) by replacing two amino acids P219 and S254 in the vicinity of the retinal with T219 and A254 correspondingly. The absorption maximum of a wild type KR2 is observed at a relatively short wavelength 525 nm. λ_{max} is an important parameter of a rhodopsin to be an optogenetic tool. In particular, red-shifted optogenetic tools are necessary to excite neurons in deeper tissues in vivo applications. The main aim of this work was to engineer a red-shifted outward sodium pump. A central idea was to probe with mutations of the retinal environment near the β -ionine ring since the mutation in the region of the Schiff base of retinal may influence functional parameters of the protein. The authors showed that a double P219T and S254A mutation resulted in the required shift. The light-driven sodium pump rhodopsin is a potential optogenetic tool therefore this result may lead to future useful applications. In general, this work comprises useful information for color tuning of other rhodopsins.

“Red-shifting Mutation of Light-driven Sodium Pump Rhodopsin” by Inoue et al. is a solid experimental work but it lacks sufficient novelty which is a reason for publishing it in Nature Communications. Indeed, color tuning is quite an old story in rhodopsin field. Indeed, in the introduction of J. Am. Chem. Soc. 2006, 128, 10808-10818 the authors systemize major ideas of color tuning which were already discussed in literature and experimentally proven: “Several mechanisms for color tuning in these systems have been proposed: (i) coplanarization of the ring-chain system and further distortion of the chromophore structure; (ii) electrostatic interaction of the chromophore with ionic, polar and polarizable groups of the protein environment; and (iii) a change in the interactions between the chromophore and its complex counterion.” Color tuning was also discussed in The ISME Journal (2007), 48–55, in particular, the importance of retinal environment near the β -ionine ring in color tuning was considered in detail. Already a long time ago Luecke et. al. Science 293 (2001) 1499–1503 mentioned the following “Second, a blue shift is expected from the removal of two hydroxyls near the b-ionone ring from Ser141 and Thr142 in BR, replaced with nonpolar residues Gly130 and Ala131 in NpSRII.” In the manuscript by Inoue et al. the introduction of two hydroxyls in the vicinity of the β -ionine ring was used to obtain a red-shifted sodium pump. Such an inverse (in comparison with Luecke et al.) mutation is straightforwardly evident and cannot be considered a new approach. I have a feeling that the authors themselves recognize this. They describe in a long introduction some of the known published facts which helped them rationalize screening of the mutations to obtain a red-shifted mutant. Nevertheless this work is useful since it specifies proper mutations and may be a useful technical achievement with a detailed biophysical characterization of the studied mutants and deserves to be published, for instance, in Scientific Reports or Biophys J.

Reviewer #3 (Remarks to the Author):

Kandori and co-workers have designed a redshifting variant of the sodium-pumping rhodopsin KR2. Based on substitutions found in natural bacterial rhodopsin, the authors screened different mutations based on which a double variant (P219T/S254A) was created, with a 40 nm-redshifted dark state. The variant was characterized by UV-VIS & FTIR experiments, pumping measurements, and QM/MM calculations. I find that the work could be of potential interest for Nature Comms, but first some aspects needs to be clarified:

1. Abstract: ARM is not a standard abbreviation. Please reformulate

2. Abstract: what is meant by the statement that the calculations gave both qualitative and quantitative explanations? Is not one contained in the other?

3. p2 line 42: please reformulate: "and so on".

4. p.3 line 71: please provide a reference that S1 is more sensitive to the charge perturbations.

5. p. 3: there is a discussion about tuning KR2 by electrostatic effects, and the difficulty in manipulating the shift by strain effects. However, the discussion section (p. 14) suggest that the introduced mutations highly distort the retinal. This would be expected to lead to a strain induced tuning effect. A few sentences later, however, the author refer to dipole effects. The authors should clarify the tuning mechanism and link these to, e.g., computed strain and electrostatic effects. A clear figure on the structural model with the distorted retinal environment could also help. In this context, the authors should also better refer to previous literature on electrostatic/tuning mechanisms.

6. p8, line 194 the QM/MM models are introduced out of the blue. Please properly introduce the calculations.

7. p 11, line 268: removal of point charges without relaxing cannot provide a quantitative estimate of the tuning effects.

8. Table 1. Most protein excitation energies seem to be in good agreement with experiments. However, what is the reason for the larger discrepancy for P219G between the computed (526 nm) and experimental value (542 nm)?

9. Table 1: The predicted vacuum retinal excitation energies are in the 722-775 nm range. Retinal in vacuum is expected to absorb in the 550-600 nm range. What is the reason for this large discrepancy?

10. I do not agree on the explanation of the shifts given Fig. 6: A positive dipole is drawn from the oxygen of the hydroxide to the beta-ionine unit. However, the oxygen is more electronegative (e.g. -0.66e in the CHARMM force field) than the beta-ionone unit. The more plausible explanation is that the positive charge on the beta-ionone in S1 is stabilized by the negative charge on the Thr unit. Please comment and revise model and discussion accordingly.

11. Is the amount of charge that moves to the beta-ionone unit the same in WT and all mutants?

12. p. 17, line 425: Please reformulate: "synthesis of the QM/MM models"

13. p. 18, line 426: how was the membrane-water-ion environment modeled? Please comment on how the authors expect that the surroundings affect the predicted tuning effects?

14. p. 30 figure 1: please clearly indicate the location of S254.

15. p. 34: figure 5b is difficult to understand. Please move this to the SI. This data should also be plotted rather than listing so many values.

16. p. 35, the low quality of figure makes it very difficult to read (please also see my comment above). Please revise the figure accordingly.

17. There seems to be a recent QM/MM study on optical spectra in KR2: PNAS 114 7043-7048, 2017. This work should be cited.

Replies to the comments of Reviewer 1

Reviewer #1 (Remarks to the Author):

The authors report on the engineering of a red-shifted Na rhodopsin. Super important as a tool for optogenetics.

Dear Dr. Oded Béjà,

Thank you for your quite positive comments (*italic black*) on our work. We revised the manuscript according to your comments.

Remarks:

The first sentence of the introduction should be in plural. RhodopsinS.

We appreciate your careful reading and the manuscript was revised according to your suggestion.

It would be extremely important to mark the P219 S254 positions in figure S1. And to also include it that figure the sequence of JsNaR.

Oded Beja

Thank you for the fruitful suggestion. We marked those residues and included the sequence of JsNaR in the Supplementary Figure 1 in the revised manuscript as follows.

Supplementary Figure 1. Multiple alignment of amino acid sequences of microbial rhodopsins. The amino acid sequences were aligned by ClustalW¹ for BR, BR of *Haloquadratum walsbyi* (*HwBR*), ChR2 without the C-terminal sequence, Chrimson, KR2 and *JsNaR*. The residue numbers (Res. No.) in BR and KR2 are shown in the first and second row, respectively. The positions of KR2 Pro219 and Ser254 are indicated by pink and orange diamonds, respectively.

Replies to the comments of Reviewer 2

Reviewer #2 (Remarks to the Author):

The manuscript “Red-shifting Mutation of Light-driven Sodium Pump Rhodopsin” by Inoue et al. is describing a 40-nm red-shift of the wavelength of the maximum of absorption (λ_{max}) of a sodium pump rhodopsin (KR2) by replacing two amino acids P219 and S254 in the vicinity of the retinal with T219 and A254 correspondingly. The absorption maximum of a wild type KR2 is observed at a relatively short wavelength 525 nm. λ_{max} is an important parameter of a rhodopsin to be an optogenetic tool. In particular, red-shifted optogenetic tools are necessary to excite neurons in deeper tissues in vivo applications. The main aim of this work was to engineer a red-shifted outward sodium pump. A central idea was to probe with mutations of the retinal environment near the β -ionine ring since the mutation in the region of the Schiff base of retinal may influence functional parameters of the protein. The authors showed that a double P219T and S254A mutation resulted in the required shift.

We thank the reviewers for his/her detailed review and positive comments (*italic black*) that,

“The light-driven sodium pump rhodopsin is a potential optogenetic tool therefore this result may lead to future useful applications. In general, this work comprises useful information for color tuning of other rhodopsins.”

and for his/her criticism,

“Red-shifting Mutation of Light-driven Sodium Pump Rhodopsin” by Inoue et al. is a solid experimental work but it lacks sufficient novelty which is a reason for publishing it in Nature Communications. Indeed, color tuning is quite an old story in rhodopsin field. Indeed, in the introduction of J. Am. Chem. Soc. 2006, 128, 10808-10818 the authors systemize major ideas of color tuning which were already discussed in literature and experimentally proven: “Several mechanisms for color tuning in these systems have been proposed: (i) coplanarization of the ring-chain system and further distortion of the chromophore structure; (ii) electrostatic interaction of the chromophore with ionic, polar and polarizable groups of the protein environment; and (iii) a change in the interactions between the chromophore and its complex counterion.” Color tuning was also discussed in The ISME Journal (2007), 48–55, in particular, the importance of retinal environment near the β -ionine ring in color tuning was considered in detail. Already a long time ago Luecke et al. Science 293 (2001) 1499–1503 mentioned the following “Second, a blue shift is expected from the removal of two hydroxyls near the β -ionone ring from Ser141 and Thr142 in BR, replaced with nonpolar residues Gly130 and Ala131 in NpSRIL.” In the manuscript by Inoue et al. the introduction of two hydroxyls in the vicinity of the β -ionine ring was used to obtain a red-shifted sodium pump. Such an inverse (in comparison with Luecke et al.) mutation is straightforwardly evident and cannot be considered a new approach. I have a feeling that the authors themselves recognize this. They describe in a long introduction some of the known published facts which helped them rationalize screening of the mutations to obtain a

red-shifted mutant. Nevertheless this work is useful since it specifies proper mutations and may be a useful technical achievement with a detailed biophysical characterization of the studied mutants and deserves to be published, for instance, in Scientific Reports or Biophys J.

As this reviewer suggested there are many earlier researches on the color tuning rule of rhodopsins. However, we consider that the most outstanding finding in our work is “color tuning without impairing the biological function”. Although the earlier works suggested by the reviewer focused on the color tuning mechanism of rhodopsins (except for the work in The ISME Journal (2007) which investigated geographical distribution of green and blue absorbing proteorhodopsins and is not related to the molecular mechanism of color tuning), they were not concerned about the functional alteration by mutations. Both the P219T and S254A mutations reported in our work significantly shift the absorption while retaining transport as efficient as that of the wildtype. We believe that the discovered mutations are applicable to many other types of ion-transporting rhodopsins to develop new optogenetic tools, and difficult to predict based on the insights reported before. In addition, we revealed that the mutation of Pro219 is naturally occurring in *Jannaschia seosinensis* to adapt favorable sun-light wavelength without loss of the function. Therefore, we believe this work would attract broad interest from not only biophysical but also optogenetics, microbial, evolutionary researchers. In order to more clearly show this point, we added the following sentence in the last paragraph of the manuscript:

Page 17, line 406-410

The molecular-level design of rhodopsins regulating the absorption wavelength of the retinal chromophore without impairing the transport activity is still difficult compared to the prediction of the only absorption wavelength. Thus, our findings expand the experimental basis useful to establish the artificial design of functional molecules useful for optogenetics application.

Also, since we consider the previous works mentioned by the reviewer would be insightful for the readers, we cited these works in the revised manuscript:

Page 4-5, line 100-102

The O-H bearing Ser141 and Thr142 neighbouring to the β -ionone ring in BR were also suggested to contribute to the red-shift of the absorption by structural, mutational and theoretical studies³⁸⁻⁴⁰.

Page 16, line 387-388

“L/Q switch”, is well known to regulate the spectra between the green- and blue-absorbing types, respectively.⁵⁴⁻⁵⁶

References

38 Luecke, H., Schobert, B., Lanyi, J. K., Spudich, E. N. & Spudich, J. L. Crystal structure of sensory rhodopsin II at 2.4 angstroms: Insights into color tuning and transducer interaction. *Science* **293**,

1499-1503 (2001).

40 _____ Hoffmann, M. *et al.* Color tuning in rhodopsins: The mechanism for the spectral shift between bacteriorhodopsin and sensory rhodopsin ii. *J. Am. Chem. Soc.* **128**, 10808-10818 (2006).

56 _____ Sabehi, G. *et al.* Adaptation and spectral tuning in divergent marine proteorhodopsins from the eastern mediterranean and the sargasso seas. *ISME J.* **1**, 48-55 (2007).

Replies to the comments of Reviewer 3

As we report in the following, the reviewer comments (*italic black*) have been carefully considered and the corresponding point-to-point answers are given below. Below, we also report on the corresponding changes to the main text and supporting information that, we believe, should satisfy the reviewer's requests.

Reviewer #3 (Remarks to the Author):

Kandori and co-workers have designed a redshifting variant of the sodium-pumping rhodopsin KR2. Based on substitutions found in natural bacterial rhodopsin, the authors screened different mutations based on which a double variant (P219T/S254A) was created, with a 40 nm-redshifted dark state. The variant was characterized by UV-VIS & FTIR experiments, pumping measurements, and QM/MM calculations. I find that the work could be of potential interest for Nature Comms, but first some aspects needs to be clarified:

We thank the reviewers for his/her detailed review and positive comments (*italic black*) that,

- 1. Abstract: ARM is not a standard abbreviation. Please reformulate*
- 2. Abstract: what is meant by the statement that the calculations gave both qualitative and quantitative explanations? Is not one contained in the other?*

We are grateful to the reviewer for his/her remark and questions. We agree with the reviewer that the ARM label is non-standard and that the sentence referring to a qualitative and quantitative explanation is confusing and needs to be changed. With regard to this second point, we meant that, as documented and reported in the present manuscript, the type of QM/MM models used in our research can reproduce the observed trend in wavelength of the absorption maxima and, in turn, can be used as an analytical tool to evaluate, for each mutation, the magnitude of the electrostatic effect associated with each given amino acid replacement.

In order to make the abstract more readable and avoid confusion, the original sentence:

“...QM/MM models generated with ARM protocol showed qualitatively and quantitatively that the changes in the electrostatic interaction between retinal and the mutated residues lowered the energy gap between the ground and the first electronically excited state....”

has been replaced with the following sentence:

Page 2, line 34-37

“...QM/MM models generated with an automated protocol showed that the changes in the electrostatic interaction between protein and retinal chromophore induced by the amino acid replacements, lowered the

energy gap between the ground and the first electronically excited state....”

3. p.2 line 42: please reformulate: "and so on".

Thank you for this suggestion. The sentence has been changed to:

Page 2, line 42-43

“...Microbial rhodopsins are photoreceptive membrane proteins widely distributed among diverse eubacteria, archaea, eukaryotic algae, fungi and giant viruses.^{1,2}...”

4. p.3 line 71: please provide a reference that S_1 is more sensitive to the charge perturbations.

The original sentence led to a misinterpretation. In fact, we are not aware of any study showing that the S_1 , rather than the S_0 state, of the retinal chromophore is more sensitive to an external charge perturbation. We just wanted to stress the fact (discussed in refs. 28 and 29 and in the new ref. 30 - notice that ref. 29 in the revised manuscript is the original ref. 60: PNAS (2006), 103, 17154-17159 and ref. 30 is a new reference: Chem. Rev. (2017), 117, 13502-13565) that the positive charge of the retinal chromophore is located in different “regions” in the S_0 and S_1 states, namely near the C=N bond and near the β -ionone ring respectively.

For sake of clarity, the original sentence:

“... Specifically, a negative charge near the Schiff-base region selectively stabilizes the S_0 state, whereas the energy level of the S_1 state is more sensitive to the charge near the β -ionone ring...”

has been replaced with the following sentences:

Page 3, line 69-70

“...While this has a localized positive charge on the Schiff-base linkage in the S_0 state, it delocalizes in the S_1 state toward the β -ionone ring.^{28-30,}”

Page 3, line 72-74

“... Specifically, a negative charge near the Schiff-base region selectively stabilizes the S_0 state, whereas the energy level of the S_1 state is stabilized by a negative charge located near the β -ionone ring³¹...”

The following reference has been repositioned:

The original reference 60 (PNAS (2006), 103, 17154-17159) → Now is reference 29

A new reference has been added:

New reference 30: Gozem, S.; Luk, H. L.; Schapiro, I.; Olivucci, M. Theory and Simulation of the Ultrafast Double-Bond Isomerization of Biological Chromophores. *Chem. Rev.* **117**, 13502-13565 (2017).

5. p. 3: *there is a discussion about tuning KR2 by electrostatic effects, and the difficulty in manipulating the shift by strain effects. However, the discussion section (p. 14) suggest that the introduced mutations highly distort the retinal. This would be expected to lead to a strain induced tuning effect. A few sentences later, however, the author refer to dipole effects. The authors should clarify the tuning mechanism and link these to, e.g., computed strain and electrostatic effects. A clear figure on the structural model with the distorted retinal environment could also help. In this context, the authors should also better refer to previous literature on electrostatic/tuning mechanisms.*

We have considered the point raised by the reviewer and clarified it. In fact, the considered mutations does distort the chromophore with respect to the WT geometry. On the other hand, we find that, as stressed in the manuscript, such distortions blue-shift the absorption. However, the electrostatic interactions introduced by the same mutation red-shift it at a larger and dominating extent.

The following parts have been improved for clarifying the matter:

Page 11, line 259-263

“...The constructed QM/MM models allowed to disentangle the electrostatic and steric effects responsible for the observed ΔE_{S1-S0} red-shifted values. To do so we computed the ΔE_{S1-S0} values of the retinal chromophore in isolated condition (i.e. removing the protein part from the model while keeping the chromophore geometry fixed at its equilibrium geometry in the protein environment)...”

Page 11, line 382-387

“...Interestingly, it was found that the mutation-induced changes in the retinal geometry cause a blue-shift with respect to the WT. However, as shown in Table 1 (right column), we also found that a red-shifting electrostatic contribution imposed by the protein environment dominates, thus quenching the chromophore steric effect and resulting in a net red-shifted change, with the strongest effect observed for the P219T/S254A mutant...”

6. p8, line 194 *the QM/MM models are introduced out of the blue. Please properly introduce the calculations.*

Many thanks to the reviewer for identifying this ambiguity. In order to clarify the point above, the original sentence in section “Spectroscopic Analysis of KR2 Mutants”:

“... we firstly investigated the light-induced structure difference of the retinal by light-induced difference Fourier transform infrared (FTIR) spectroscopy. The whole spectra ...”

has been replaced with the following sentence:

Page 8, line 188-191

“...Fourier transform infrared (FTIR) spectroscopy measurements and support such an investigation by building and analyzing quantum mechanics/molecular mechanics (QM/MM) models of wild type KR2 and its mutants (see section “QM/MM Model of KR2 WT and Mutants”). The whole spectra ...”

7. p 11, line 268: removal of point charges without relaxing cannot provide a quantitative estimate of the tuning effects.

The idea is to analyze the tuning associate to a mutation by hypothesizing that it can be seen as a sum of electrostatic (point charges) and steric (chromophore and cavity geometrical distortion). We determine the electrostatic effect at the mutant equilibrium geometry by switching off the protein charges to isolate the geometrical effect.

In order to clarify the meaning of our analysis and avoid confusion, the original sentence:

“... In addition, using the QM/MM models, a quantitative explanation regarding the red-shift effect observed in the mutants, with respect to the WT, can be provided. We can evaluate the specific effect of each mutation, by setting the point charges of the side-chain under investigation to zero and then use the same QM/MM models to re-compute the vertical excitation energy (ΔE_{off})....”

has been replaced with the following sentence:

Page 12, line 281-283

“...Using the same QM/MM models, it is also possible to investigate the role played by each protein amino acid residues in the described red-shifting relative to the WT form. In fact, one can set the point charges of each specific residue to zero and then re-compute the ΔE_{S1-S0} value (ΔE_{off})....”

8. Table 1. Most protein excitation energies seem to be in good agreement with experiments. However, what is the reason for the larger discrepancy for P219G between the computed (526 nm) and experimental value (542 nm)?

The experimental value of KR2 P219G is 535 nm (Page 13, line 309 in the original manuscript). Therefore the amount of the shift from the λ_{max} of KR2 WT is reproduced reasonably well by the calculation. In the original manuscript, we claimed that the spectrum of KR2 P219G is shown in Fig. 7a. However, due to an error in the production of the figure, that was not the case. Therefore, we added the spectrum of KR2 P219G into Fig. 7a.

Figure caption of Figure 7a

The absorption spectrum of KR2 wildtype and P219G mutant shown by the magenta dotted line and cyan solid line, respectively.

Revised Figure 7

9. Table 1: The predicted vacuum retinal excitation energies are in the 722-775 nm range. Retinal in vacuum is expected to absorb in the 550-600 nm range. What is the reason for this large discrepancy?

We thank the reviewer for his/her remark. We agree that the all-*trans* chromophore in *vacuum* and in its equilibrium geometry absorbs around 600 nm. Such observed absorption (see for instance Andersen, L. H. et al. J. Am. Chem. Soc. 2005, 127, 12347-12350) has been reproduced using the same level of theory used in our paper (see for instance Cembran, A. et al. J. Phys. Chem. A 2005, 109, 6597-6605 and Rajput, J. et al. Angew. Chem. Inter. Ed. 2010, 49, 1790-1793). After we double checked the vertical excitation energies computed in *Vacuum* in Table 1 of the original manuscript, we discovered a typographical error in compiling the table. In fact, the correct computed red-shifted excitation energy values are only 6-12 kcal/mol red-shifted with respect to the protein values and not 12-20 kcal/mol as originally reported. Therefore, we modified Table 1 in the main text which now includes the correct values yielding wavelengths around 620 nm due to the fact that the in *Vacuum* chromophores are not relaxed being taken with their protein optimized geometries.

Revised Table 1

Table 1. The energy differences ($\Delta E_{S_1-S_0}$) between the ground (S_0) and first electronically excited state (S_1) were calculated by the QM/MM models using Automatic Rhodopsin Modeling (ARM). The values for the retinal in the protein (Protein), isolated in vacuum (Vacuum), and their difference (Protein - Vacuum) are shown. The values in the parenthesis for the mutants show the difference from KR2 WT.

Protein	$\Delta E_{S_1-S_0}$ (Protein) (kcal/mol)	$\Delta E_{S_1-S_0}$ (Vacuum) (kcal/mol)	$\Delta E_{S_1-S_0}$ (Protein - Vacuum) (kcal/mol)
KR2 WT	55.2	43.1	+12.1
P219G	54.3 (-0.9)	43.8 (+0.7)	+10.5 (-1.6)
P219T	53.5 (-1.7)	44.5 (+1.3)	+9.0 (-3.1)
S254A	53.1 (-2.1)	43.6 (+0.5)	+9.5 (-2.6)
P219T/S254A	51.5 (-3.7)	45.9 (+2.7)	+5.6 (-6.5)

10. I do not agree on the explanation of the shifts given Fig. 6: A positive dipole is drawn from the oxygen of the hydroxide to the beta-ionone unit. However, the oxygen is more electronegative (e.g. $-0.66e$ in the CHARMM force field) than the beta-ionone unit. The more plausible explanation is that the positive charge on the beta-ionone in S_1 is stabilized by the negative charge on the Thr unit. Please comment and revise model and discussion accordingly.

We are indebted to the reviewer for his/her very helpful comment. Indeed, regrettably, the interpretation of the generated dipole was not correct for P219T and P219T/S254A. Following the suggestion in point 16 of the reviewer criticism, we have now modified Figure 6 to provide a better and qualitatively correct visualization of the retinal and residues and we increased the quality of the figure. The new version of Figure 6 is reported below (see point 16 for the inserted changes).

11. Is the amount of charge that moves to the beta-ionone unit the same in WT and all mutants?

We analysed the total positive charge on the C11H-C12H-C13(Me)-C14H-C15H-NHR moiety (i.e., relevant to the C11=C12 isomerization) of the retinal chromophore for the ground (S_0), first (S_1) and second (S_2) singlet states. Also, we performed the same analysis for the charges corresponding to the shorter C13(Me)-C14H-C15H-NHR moiety (i.e., relevant to the C13=C14 isomerization, see values in parenthesis). The corresponding data show that there is not a large difference in the charge movement between KR2 WT and its mutants. These data have now been inserted in the Supporting Information file as

Supplementary Table 5.

Supplementary Table 5. Total positive charge on the C11H-C12H-C13(Me)-C14H-C15H-NHR moiety (i.e., relevant to the C11=C12 isomerization). In *parenthesis* we also give the charges corresponding to the shorter C13(Me)-C14H-C15H-NHR moiety (i.e., relevant to the C13=C14 isomerization).

Protein	S₀ Charge (e)	S₁ Charge (e)	S₂ Charge (e)
KR2 WT	0.83 (0.78)	0.45 (0.42)	0.74 (0.70)
P219G	0.85 (0.80)	0.42 (0.39)	0.76 (0.72)
P219T	0.84 (0.78)	0.44 (0.40)	0.74 (0.70)
S254A	0.87 (0.81)	0.41 (0.37)	0.79 (0.73)
P219T/S254A	0.81 (0.77)	0.47 (0.45)	0.72 (0.69)

12. p. 17, line 425: Please reformulate: "synthesis of the QM/MM models"

Thanks the reviewer for detecting this inconsistency/typo. We modified the corresponding sentence in the main text as follows:

Page 18-19, line 449-452

"...As anticipated above, the QM/MM models of the KR2 WT and its mutants P219T, P219G, S254A and P219T/S254A have been constructed employing the ARM protocol which has been benchmarked with respect to a set of animal and microbial rhodopsins from different organisms and a set of mutants from bovine rhodopsin..."

13. p. 18, line 426: how was the membrane-water-ion environment modeled? Please comment on how the authors expect that the surroundings affect the predicted tuning effects?

As we described in the original ref. 46 (Melaccio *et al.* (2016)), the QM/MM models constructed with the ARM protocol are a gas-phase and globally uncharged monomer protein models, which do not incorporate the protein environment (membrane and solvent). As demonstrated in the mentioned paper, the effect of the protein environment is indirectly incorporated with the introduction of Cl⁻ and Na⁺ counterions on the cytoplasmic and external faces of the rhodopsin structure. Of course, such only apparently basic models have been benchmarked. In fact, it has been shown that they can reproduce the observed trends in excitation energies with, most frequently, a systematic blue shifted error of 2-3 kcal/mol.

In order to clarify the meaning of our analysis and avoid confusion, we have improved the following text in the Methods section:

“...While ARM models are basic gas-phase and globally uncharged monomer models constructed starting from an X-ray crystallographic structure or comparative model, the benchmark shows that ARM models reproduce the experimental λ_{\max} values with a few kcal/mol discrepancy and that can reproduce trends in λ_{\max} . Accordingly, in the present contribution, the rhodopsin models are used to study the molecular mechanics determining the λ_{\max} changes after having been validated by reproducing the observed λ_{\max} ...”

14. p. 30 figure 1: please clearly indicate the location of S254.

We now show Ser254 in Figure 1 and refer to it in the figure legend:

Revised Figure 1

Figure 1. Non-polar amino acid residues around the retinylidene β -ionone ring in KR2 replaced with polar ones in this study. The residues mutated to the identical ones as occurring in Chrimson and nine further screened residues are coloured in orange and green, respectively. The C α atoms are shown as spheres for Gly residues. Ser254 near the retinylidene moiety (in yellow) is coloured in cyan.

15. p. 34: figure 5b is difficult to understand. Please move this to the SI. This data should also be plotted rather than listing so many values.

We thank the reviewer for suggesting how to improve Figure 5. We decided to modified Figure 5 in the following way: (i) Figure 5a remains as previously presented, (ii) we created two new figures, Figure 5b and 5c, in which we show, for each mutant, the difference in bond lengths and dihedral angles relative to the corresponding KR2 WT values. The corresponding numerical data are then collected in two new supplementary, Tables 3 and 4, in the Supporting Information.

Notice that we also improved the quality of the figures (600 DPI) and the absolute width for a single-column figure of 1040 pixels wide. The new supplementary Tables 3 and 4 and the modified Figure 5 are reported below:

Supplementary Table 3. Double and single bond lengths (Å) of QM/MM models built with ARM protocol at CASSCF(12,12)/6-31G*/AMBER level of theory of KR2 WT and its mutants.

Bond/Sample	WT	P219T	S254A	P219T/S254A
C5=C6	1.37	1.371	1.371	1.375
C6-C7	1.486	1.484	1.483	1.481
C7=C8	1.355	1.356	1.355	1.356
C8-C9	1.475	1.473	1.472	1.473
C9=C10	1.362	1.362	1.362	1.365
C10-C11	1.456	1.450	1.453	1.451
C11=C12	1.355	1.357	1.357	1.360
C12-C13	1.462	1.461	1.464	1.465
C13=C14	1.362	1.364	1.368	1.369
C14-C15	1.445	1.442	1.439	1.439
C15=N	1.289	1.288	1.288	1.290

Supplementary Table 4. Double and single bond lengths (Å) of QM/MM models built with ARM protocol at CASSCF(12,12)/6-31G*/AMBER level of theory of KR2 WT and its mutants.

Dihedral/Sample	WT	P219T	S254A	P219T/S254A
C4-C5=C6-C7	172.46	171.92	172.50	171.49
C5=C6-C7=C8	-175.74	-174.40	-174.26	-175.44
C6-C7=C8-C9	173.97	173.45	173.98	173.14
C7=C8-C9=C10	-172.54	-173.27	-172.04	-173.40
C8-C9=C10-C11	170.81	170.76	171.61	170.12
C9=C10-C11=C12	-178.55	-177.45	-178.93	-177.27

C10-C11=C12-C13	162.81	162.35	161.32	160.18
C11=C12-C13=C14	-171.66	-172.00	-170.36	-169.90
C12-C13=C14-C15	151.20	151.39	150.02	149.41
C13=C14-C15=N	-178.92	-177.98	-176.10	-174.46
C14-C15=N-C _δ	159.55	159.86	157.89	152.10

Figure 5. Excitation energy and conjugated structure of retinal chromophore computed by QM/MM models. (a) Comparison between the computed and observed vertical excitation energies, ΔE_{S1-S0} (kcal/mol) of QM/MM models build with ARM protocol at CASPT2//CASSCF(12,12)/6-31G*/AMBER level of theory, for KR2 WT (dark blue), P219G (pink), P219T (clear blue), S254A (green), and P219T/S254A (red) mutants. The error bars of the standard deviation are show in black (see details in Supplementary Table 1). (b) Bond lengths and (c) chromophore dihedral angle differences of each mutant relative to KR2 WT at CASSCF(12,12)/6-31G*/AMBER level of theory.

16. p. 35, the low quality of figure makes it very difficult to read (please also see my comment above). Please revise the figure accordingly.

We thank the reviewer for his/her remark. We improved the quality of the figure producing a version with a 600 DPI resolution and the absolute width for a double-column figure of 2080 pixels wide. The modified Figure 6 is reported below:

Figure 6. QM/MM structures around retinal chromophore in KR2 WT and mutants. Comparison between retinal chromophores and mutated residues 219 and 254 in (a) KR2 WT, (b) P219G, (c) P219T, (d) S254A and (e) P219T/S254A mutants. For mutants are also shown, in transparent representation, the retinal chromophore and 219 and 254 residues of KR2 WT.

17. There seems to be a recent QM/MM study on optical spectra in KR2: PNAS 114 7043-7048, 2017. This work should be cited.

We thank the reviewer for pointing out this literature omission. We now introduce the recommended reference at the end of the following period:

Page 19, line 458-461

“... Thus, ARM does not reproduce the most accurate QM/MM models possible, but computationally fast models for the rationalization of trends between sequence variability and function. We notice that, a more realistic but computationally demanding QM/MM model of KR2 has been recently reported.⁵⁹...”

The reference introduced is:

Suomivuori, C. M., Gamiz-Hernandez, A. P., Sundholm, D., & Kaila, V. R. Energetics and dynamics of a light-driven sodium-pumping rhodopsin. *Proc. Natl. Acad. Sci. USA* **114**, 7043-7048 (2017)

REVIEWERS' COMMENTS:

Reviewer #3 (Remarks to the Author):

My previous reviewer comments have been properly answered. The corrected tuning model seems now consistent with the expected physical picture and observed results, and the clarifications made in the text have helped to avoid misunderstandings. I find that the achieved tuning effect, while retaining the biological activity of KR2, and additionally providing a semi-quantitative explanation for the effect, makes an important contribution to the field. I therefore recommend the manuscript for publication in NComms.